# Community pharmacy and general practice collaborative and integrated working: a realist review protocol

Emily Claire Owen ,[1] Ruth Abrams,[2] Ziyue Cai,[3] Claire Duddy ,[4]
Nina Fudge ,[5] Julia Hamer-Hunt,[1,6] Fran Husson,[1,7] Kamal Ram Mahtani,[4]
Margaret Ogden,[1] Deborah Swinglehurst ,[5] Malcolm Turner,[1] Cate Whittlesea,[3]
Geoff Wong,[4] Sophie Park [1]

¹Department of Primary Care & Population Health, University College London, London, UK
²School of Health Sciences, University of Surrey, Surrey, UK
³Research Department of Practice and Policy, School of Pharmacy, University College London, London, UK
⁴Nuffield Department of Primary Care Health Sciences, University of Oxford, Oxford, UK
⁵Wolfson Institute of Population Health, Queen Mary University of London, London, UK, London, UK
⁶Department of Psychiatry, University of Oxford, Oxford, UK
⁷Faculty of Medicine, Imperial College London, London, UK

**Correspondence to**
Dr Emily Claire Owen;
emily.owen@ucl.ac.uk

## ABSTRACT

**Introduction** Increasing collaborative and integrated working between General practice (GP) and Community pharmacy (CP) is a key priority of the UK National Health Service and has been proposed as a solution to reducing health system fragmentation, improving synergies and coordination of care. However, there is limited understanding regarding how and under which circumstances collaborative and integrated working between GP and CP can be achieved in practice and how regulatory, organisational and systemic barriers can be overcome.

**Methods and analysis** The aim of our review is to understand how, when and why working arrangements between GP and CP can provide the conditions necessary for optimal communication, decision-making, and collaborative and integrated working. A realist review approach will be used to synthesise the evidence to make sense of the complexities inherent in the working relationships between GP and CP. Our review will follow Pawson's five iterative stages: (1) finding existing theories; (2) searching for evidence (our main searches were conducted in April 2022); (3) article selection; (4) data extraction and (5) synthesising evidence and drawing conclusions. We will synthesise evidence from grey literature, qualitative, quantitative and mixed-methods research. The research team will work closely with key stakeholders and include patient and public involvement and engagement throughout the review process to refine the focus of the review and the programme theory. Collectively, our refined programme theory will explain how collaborative and integrated working between GP and CP works (or not), for whom, how and under which circumstances.

**Ethics and dissemination** Formal ethical approval is not required for this review as it draws on secondary data from published articles and grey literature. Findings will be widely disseminated through: publication in peer-reviewed journals, seminars, international conference presentations, patients' association channels, social media, symposia and user-friendly summaries.

**PROSPERO registration number** CRD42022314280.

## STRENGTHS AND LIMITATIONS OF THIS STUDY

⇒ This is the first realist review to explore collaborative and integrated working between general practice and community pharmacy.
⇒ Public contributor coapplicants and key stakeholders will contribute to the development and refinement of the programme theory, analysis, interpretation, dissemination of findings and help to ensure our research informs future practice.
⇒ The review findings may be transferable to other primary care interfaces and the future productive shaping of integrated and collaborative working.
⇒ Our review may be limited by the quality and relevance of existing literature in this field.
⇒ We will only include literature that is written or translated into English.

## INTRODUCTION

The National Health Service (NHS) Long-Term Plan[1] sets out fundamental changes to the nature and provision of UK primary care. This includes the introduction of 'integrated care systems' (ICSs) and legislative changes to support collaborative and integrated working between individuals and institutional organisations. It recommends significant expansion in the numbers of allied healthcare professionals working in primary care settings, and related extension in their roles and responsibilities. For instance, the role of community pharmacy (CP) has evolved from supporting general practice (GP) (eg, medicines use review) towards joint working.[2 3]

There are many different definitions used in the healthcare literature to describe collaborative and integrated working between health professionals. Some definitions indicate that collaboration involves shared goals and responsibility, decision-making, trust, and open, honest communication.[4] Integration may include common policies and incentives, defined referral mechanisms, practice guidelines and formal structures.[4 5] Effective collaborative and integrated working between GP and CP may have the potential to enhance access to services, improve interprofessional communication, and continuity of patient

care.[6] Policy direction and expectations are clear: shifting away from competition towards more collaborative ways of working.[7] Examples include recent National Institute for Health and Care Excellence standards to facilitate effective partnership working,[8] 'influenza vaccine principles,[9] and enhanced service amendments[10] for GP referrals to CP. However, implementing these changes has created unanticipated and unpredictable consequences. For instance, the more clinical activity is distributed across providers, healthcare disciplines and settings, the greater the risk for silo working, fragmentation of accountability and patient care.[11] To enable effective collaborative and integrated working, there remains a need to develop an in-depth understanding about how these providers work together; what influences interactions and how these impact patient trust, experience, equity and outcomes.

UK reports in the last decade consistently position CP as a solution for an overburdened GP and a means to improve access for patients from socioeconomically deprived communities, providing 'health on the high street'[12–14] including Directed Enhanced Services.[15] This has been part of the rationale behind the Long-Term Plan's goal of reorganising primary care so that CP is included in wider networks such as Primary Care Networks (PCNs) and ICSs.[1] However, implementation of collaborative working among independent contractors to the NHS, such as GP and CP, can be complex and problematic.

Professional bodies, including the Royal Pharmaceutical Society,[16] have published recommendations to help CP integrate into primary care.[17] However, these are not specific to ICSs and PCNs, and predominantly focus on tighter technology integration to facilitate communication between GP and CP. Recent guidance[9] includes expectations for integrated services, referral pathways and joint strategic needs assessments to address local health inequalities, but does not describe how these should be achieved.

Policy changes have the potential to influence how patients' access, use and experience primary care; form and sustain relationships with healthcare professionals; and navigate healthcare expertise within and between clinical encounters. This has implications for the nature of work done by professional groups including how this work is achieved, the use of technology to support working, how accountability (clinical and financial) is determined, the training and capacity building required; and importantly, how healthcare professionals work with each other to support effective, safe and equitable patient care.

Previous reviews have demonstrated that CP could deliver effective patient care in several areas, but implementation depends on successful collaboration and integration with other services.[6 18] Existing research into collaborative working in these settings found that imbalances of authority, limited understanding of others' roles and responsibilities, lack of time and financial remuneration models, and professional boundary friction when delivering patient care, can impact on the quality of care provided.[6 19 20] Collaboration and integrated working may bring new roles and ways of working for healthcare professionals in each organisation. These changes may also, however, contribute to and exacerbate ambiguity about professional roles and boundaries, potentially undermining the collaborative and integrated working between GP and CP, with resulting confusion for patients.[21] There is, therefore, a potential paradox at play, in which efforts to create greater role clarity, professional boundaries and distribution of work tasks in the service of integrating care contribute to poorly integrated patient care. Extant literature is not, however, able to explain when (in what contexts), why and how the identified factors affect working relationships between GP and CP. While current evidence indicates that encouraging GP and CP to work together is important, it is less clear how these working arrangements can be made to work well within ICS settings for both healthcare professionals and patients from diverse cultural and socioeconomic backgrounds.

## METHODS
### Review aim, questions and objectives
#### Aim
This review aims to understand how, when and why working arrangements between GP and CP can provide the conditions necessary for optimal communication, decision-making and collaborative and integrated working.

#### Research objectives
1. Develop a programme theory through an evidence synthesis of how GP and CP can optimise communication, decision-making, and collaborative and integrated working to support effective and equitable patient care.
2. Embed and respond to stakeholder and patient and public involvement and engagement (PPIE) perspectives throughout the design, analysis and report stages of the project, thus maximising the relevance and utility of review findings.
3. Make recommendations for practice and policy based on the refined programme theory.

#### Research questions
Within the existing literature, what can we learn that will help GP and CP to work together in a collaborative and integrated way to support effective and equitable healthcare outcomes? Specifically, the review will be guided by the following questions:
1. What are the mechanisms which support GP and CP to work in an integrated and collaborative way?
2. What are the important contexts which influence whether different mechanisms produce intended and unintended outcomes in GP and CP working relationships?

3. What are the interventional strategies that are likely to lead to intended and unintended outcomes within GP and CP working relationships?

## Approach

A realist review is an interpretive and theory-driven approach to synthesising evidence from grey literature (eg, policy documentation), qualitative, quantitative and mixed-methods research.[22] It enables the use of a range of data types to make sense of and address the context-sensitive outcomes arising from interactions between GP and CP. The realist approach begins by developing an initial programme theory that aims to explain how collaborative and integrated working between GP and CP may (or may not) work, and includes theorising the anticipated interactions between contexts, mechanisms and outcomes.[23] This initial programme theory is then tested (confirmed, refuted or refined) against empirical evidence throughout the review. The final phase of a realist review involves the synthesis of evidence, the formulation of context-mechanism-outcome configurations (CMOCs), and a refined programme theory which explains whether, why, how and to what extent GP and CP collaborative and integrated working practices may (or may not) support effective and equitable healthcare delivery for patients.[23] A realist review has the potential to produce meaningful, transferable findings across different structures and contexts within which GP and CP operate.

## Patient and public involvement and engagement

We will consult with stakeholders with a wide range of relevant expertise comprising: four PPIE coapplicants (including family and carer perspectives); front-line healthcare clinicians; workforce and training experts; and a multidisciplinary team of researchers. The project team (including PPIE coapplicants) will meet every 2 months, working collaboratively alongside experiential stakeholders. There will be three stakeholder meetings (lasting approximately 3 hours), including representation from small, medium and large pharmacy and medical organisations. Throughout the project, detailed notes will be made during and after each meeting, and the outcomes of our engagement with PPIE coapplicants and stakeholders will be recorded using an impact log.[24] Collectively, our PPIE coapplicants and stakeholder groups will contribute to the development and refinement of the programme theory, analysis, interpretation, dissemination activities (eg, copresenting and copublishing papers, posters, presentations and reports), and help to ensure our research informs future practice and policy in this important area of inquiry. This realist review protocol is registered in PROSPERO (reference number: CRD42022314280) and will be carried out in accordance with the quality and publication standards.[25 26]

## Step 1: finding existing theories

We will identify theories that explain how collaborative and integrated working between GP and CP is expected to 'work' to generate effective and equitable healthcare outcomes.[25] To do this, we will consult with key content experts in our stakeholder group and informally scope existing literature identified via citation tracking and snowballing,[27] which will also include identifying any formal theories that may be of relevance.[28] We will use relevant theories and data we find in this step to develop our initial programme theory, which will be further tested and scrutinised in this review. A draft of our initial programme theory is provided in figure 1.

## Step 2: searching for evidence

We will conduct systematic searches to identify a relevant 'body of literature' with which to develop and refine the programme theory from step 1. The main search strategy was designed, piloted and conducted by CD in collaboration with ECO and the project team. Our searches combined free text and subject heading (eg, MeSH) terms for GP and CP with a range of terms describing collaborative and integrated working arrangements. The full details of the search developed for MEDLINE are available in online supplemental file 1. This search strategy was translated for use in the following databases: MEDLINE (Ovid); Embase (Ovid); CINAHL (Ebsco); PsycINFO (Ovid); HMIC (Ovid); International Bibliography of the Social Sciences (ProQuest); Sociology Collection (ProQuest); the Web of Science (SSIE, SSCI, ESCI and CPCI indexes) and the King's Fund Library database. This search identified 2555 potentially relevant documents, of which 1133 were duplicates. A total of 1422 unique references were identified for screening.

At later stages of this review, we may undertake additional searching, including searches of any other relevant databases identified by CD, 'cited by' article searches, and searches of the citations contained in the reference lists of relevant documents. Additional grey literature, for example, documents produced by the Department of Health, local Clinical Commissioning Groups (or successor organisations), and pharmacists' professional groups will also be identified via searches of relevant websites. Relevant evidence including opinion and commentary will be used to inform programme theory development, along with published and unpublished evaluations of existing integrated services and collaborations which are likely to provide more detailed local contextual information.

Based on our preliminary searches, our initial inclusion criteria will include GP and CP; UK (the initial focus will be on the UK and countries with a universal healthcare system but we may draw on data from other healthcare systems); Date 2000 onwards (in order to capture literature prior to the first integrated and collaborative initiative called Medicines Use Review in 2003); and a focus on an element of the 'working relationship' between GP and CP (to include terms such as integrated and collaborative

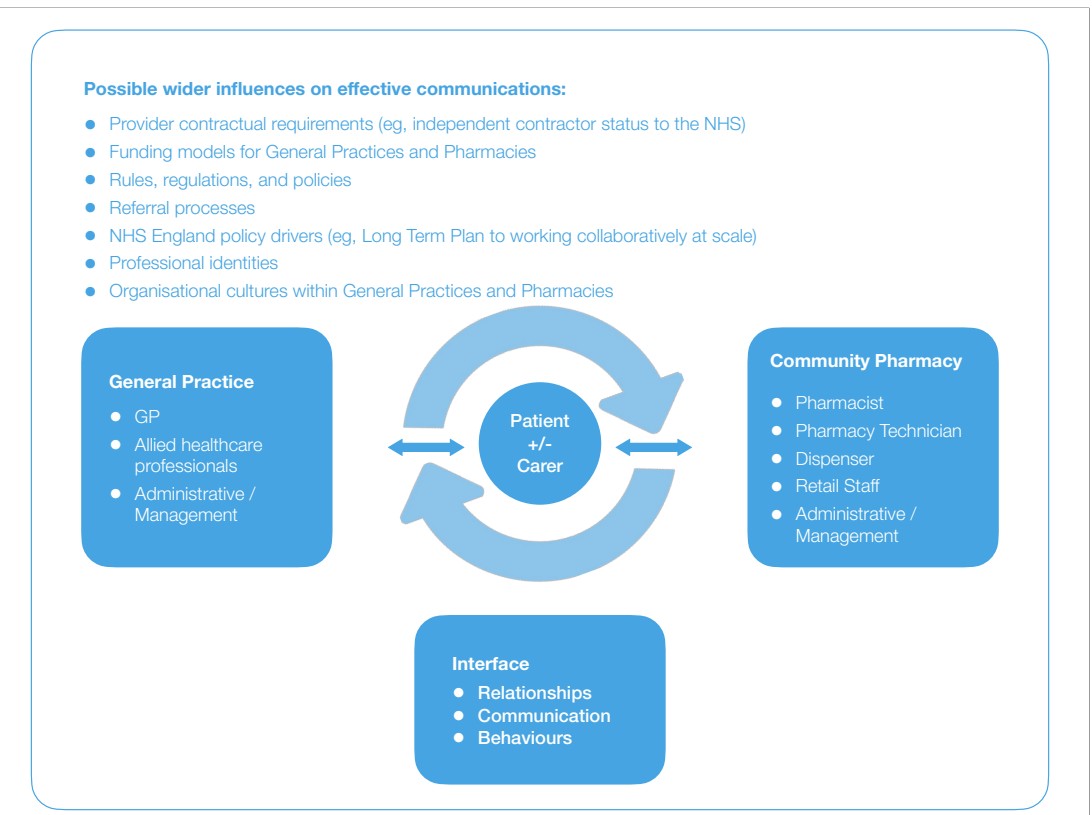

**Figure 1** Initial programme theory—used at the beginning of realist review projects to map initial explanatory theories. CP, community pharmacy; NHS, National Health Service.

working, but not exclusively). We expect to exclude papers focused entirely on clinical pharmacists working within GP settings due to differences in roles, responsibilities and the professional framework of community and clinical pharmacists.

### Step 3: article selection
This is a three-stage process: screening against title/abstract; then by full text; and finally full text documents will be selected based on their relevance (whether they contain data to contribute to theory building and/or testing) and rigour (whether the methods used to generate the relevant data are credible and trustworthy—for instance, depending on the type of document, appropriate quality standards will be used in addition to regular research team discussions).[22 25] To ensure consistency, a random 10% sample of decisions will be independently checked at each stage by CD and/or NF. Any discrepancies or disagreements will be discussed with the research team and documented.

### Step 4: data extraction
Data extraction and organisation will be undertaken by ECO. Discrepancies or disagreements will be discussed between the research team in detail and documented. The included full texts will be uploaded into qualitative data analysis software (EPPI-Reviewer Web) for coding. These will be both coded deductively (codes created in advance of data extraction and analysis, as informed by

the initial programme theory), inductively (codes created to categorise data reported in included studies), and retroductively (codes created based on an interpretation of data to infer what the hidden causal forces might be for outcomes). Each new element of relevant data will be used to refine the programme theory, and as it is refined, included studies will be rescrutinised to search for relevant data that may have been missed initially. A random sample of 10% extracted data and coding will be independently checked by CD or NF for quality control.

### Step 5: synthesising the evidence and drawing conclusions
Data analysis will use a realist logic of analysis to make sense of the initial programme theory. ECO will undertake this step with support from the research team, PPIE and stakeholders. We will use interpretive cross-case comparison to understand and explain how and why observed outcomes have occurred, for example, by comparing literature in which GP and CP have successfully worked collaboratively against those which have reported the interface as unsuccessful or detrimental, to understand how context has influenced reported findings.

We will use an established analysis and synthesis process.[28] In brief, to operationalise the realist logic of analysis, we will ask the following questions:
► Interpretation of meaning: do the documents provide data that may be interpreted as functioning as context, mechanism or outcome?

► Interpretations and judgements about CMOCs: what is the CMOC for the data that has been interpreted as functioning as context, mechanism or outcome?
► Interpretations and judgements about programme theory: how does this CMOC relate to the initial programme theory?

## ETHICS AND DISSEMINATION
### Dissemination

Our PPIE contributors and stakeholders will help us to decide on the content, storyboarding and format (eg, websites, leaflets, videos, social media). In addition to the final report, we will produce outputs directed at diverse audiences:

1. Academic outputs: for example, protocol publication. Findings (to be submitted to a high-impact, open-access and peer-reviewed journal), as well as tailored papers to different disciplinary journals.
2. Audience-specific practitioner 'how to' publications which outline practice advice on how to optimise GP and CP collaborative and integrated working for patient benefit.
3. User-friendly summaries of the review findings tailored to the needs of different audiences including the public and service users.

Our dissemination strategy will build on a participatory approach, embedding PPIE and stakeholder involvement throughout the development of this research and project timeline, including opportunities for coauthorship and copresenting. Ongoing engagement with key stakeholders will maximise opportunities to use our established networks, communication channels, and links to policy makers and providers. Our approach will be integrative, valuing the different forms of knowledge needed to produce findings capable of informing complex decision-making.[29]

### Ethics

Ethical approval is not required for this review as only secondary data sources will be used.

## DISCUSSION
### Importance of the research

This review will provide insights and recommendations to maximise GP and CP collaboration and integration. Importantly, it will help to identify where critical knowledge 'gaps' exist and propose ways to close these gaps, providing direction for future theoretical research and practice. The findings and refined programme theories will ensure patients' health and healthcare experience is central to GP and CP working relationships and processes. These working relationships and arrangements impact on patient experience, patient safety and medication errors, access, care, and formal referral, alongside professional capacity, training and workload. The review findings are likely to have broader relevance to other primary care

interfaces and the future productive shaping of collaborative and integrated working.

**Contributors** The realist review was conceptualised by SP. ECO wrote the first draft of this protocol and it was reviewed and revised by SP, CW, CD, DS, FH, GW, JH-H, KRM, MT, MO, NF, RA and ZC. CD designed, piloted and conducted the search strategy in collaboration with ECO and the project team. All authors have read and approved the final manuscript. SP is the guarantor.

**Funding** This project is funded by the National Institute for Health Research (NIHR) School for Primary Care Research (project reference 567988).

**Disclaimer** The views expressed are those of the authors and not necessarily those of the NIHR or the Department of Health and Social Care.

**Competing interests** None declared.

**Patient and public involvement** Patients and/or the public were involved in the design, or conduct, or reporting, or dissemination plans of this research. Refer to the Methods section for further details.

**Patient consent for publication** Not applicable.

**Provenance and peer review** Not commissioned; externally peer reviewed.

**ORCID iDs**
Emily Claire Owen http://orcid.org/0000-0001-5558-8567
Claire Duddy http://orcid.org/0000-0002-7083-6589
Nina Fudge http://orcid.org/0000-0002-7161-4355
Deborah Swinglehurst http://orcid.org/0000-0003-1261-9268
Sophie Park http://orcid.org/0000-0002-1521-2052

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
