## [Reviewer comments · BMJ Open]

ARTICLE DETAILS

TITLE (PROVISIONAL)	Community Pharmacy and General Practice collaborative and integrated working: a realist review protocol
AUTHORS	Owen, Emily; Whittlesea, Cate; Duddy, Claire; Swinglehurst, Deborah; Husson, Fran; Wong, Geoff; Hamer-Hunt, Julia; Mahtani, Kamal; Turner, Malcolm; Ogden, Margaret; Fudge, Nina; Abrams, Ruth; Cai, Ziyue; Park, Sophie

VERSION 1 – REVIEW

REVIEWER	Edwards, Michelle Cardiff University, Division of Population Medicine
REVIEW RETURNED	26-Oct-2022

GENERAL COMMENTS	This paper clearly sets out the rationale and methods for the realist review. I have no comments to add.
--

REVIEWER	Jeffries, Mark University of Manchester, School of Health Sciences
REVIEW RETURNED	08-Nov-2022

GENERAL COMMENTS	Community Pharmacy and General Practice collaborative and integrated working: a realist review protocol Thank you for the opportunity to review this manuscript; I very much enjoyed reading it. I think the planned review is excellent and the detail explaining how it will be undertaken is very thorough. I think the protocol itself will, and the planned review, have potential to make a significant contribution to the literature in a very important area. I have a few minor points for you to consider. I hope they are helpful. 1) In the abstract CP and GP are defined on first use but not in the main body of the manuscript, should they also be so there?2) Page 4 Line 83. I think the comment at the end of this paragraph could perhaps be strengthened by reference to more recent literature if there is any. Ref 2 is from 2003.3) Page 4 Lines 84-87. Are you referring here to primary care, secondary care or healthcare in general? Might be useful to make this clear here.4) Page 5 Lines 101-107. Useful for the rationale but would this sit better elsewhere? There is no connection to the lit here so I think for that reason it might be better placed towards the end of the background5) Page 7 Lines 150-152. Embed and respond. Isn't it more likely that you will respond to stakeholder perspectives and then embed those in all stages of the research?
--

	6) Page 7 Line 158. You have as a research question (and research objective on page 6) “to support effective and equitable healthcare outcomes”. I think that is absolutely fine. However, do you discuss health inequalities in the background? It might be worth adding something in the background to support this. It’s an important consideration because it might be that certain organisational practices or arrangements for collaborative and integrated working could potentially privilege certain groups in society so good objective but some reflection in the background might help strengthen the rationale for that RQ. 7) Page 7 Line 172. I think you need more clarity and explanation around what you mean by initial programme theory. I just think this needs a bit further explanation particularly for readers less familiar with realist approaches. 8) Apologies for being pedantic but by PPI do you mean PPIE? Is there Engagement as well as involvement? 9) In your figure outlining your initial programme theory you have GPs, all allied health professionals, and admin staff. Is it worth being a bit more specific here? GP based pharmacists and non-medical prescribers could particularly be involved in collaboration and integration between GP and CP. 10) Page 10 Lines 243-246. Article selection. Could you provide the criteria for deciding if relevant data are credible and trustworthy? 11) Page 11 Line 252 Data extraction. Discrepancies and disagreements discussed between whom? 12) Page 13 Line 305. You say you will find insights and solutions. Is it beyond the scope of a review to provide solutions? You might unpick current practice and develop it into CMOCs and your programme theory but I’m not sure if “solutions” can be offered by that? 13) Related to that can you perhaps discuss what future empirical work might be undertaken after your review? I also think some comment about the impact beyond the review and what the pathways to that impact might look like would be of use.
--	--

VERSION 1 – AUTHOR RESPONSE

Reviewer 1:	Thank you for your feedback.	
1. This paper clearly sets out the rationale and methods for the realist review. I have no comments to add.		
Reviewer 2:		
1. Thank you for the opportunity	Thank you very much for your detailed review of our manuscript.	

to review this manuscript; I very much enjoyed reading it. I think the planned review is excellent and the detail explaining how it will be undertaken is very thorough. I think the protocol itself will, and the planned review, have potential to make a significant contribution to the literature in a very important area.		
2. In the abstract CP and GP are defined on first use but not in the main body of the manuscript, should they also be there?	Thanks for pointing this out – we have now defined them in the main body of the manuscript.	L81-83 – ‘the role of Community Pharmacy (CP) has evolved from supporting General Practice (GP) (e.g., Medicines Use Review) towards joint-working).
3. Page 4 Line 83. I think the comment at the end of this paragraph could	Thank you for noticing this. We have addressed accordingly.	L83 – we have referenced more recent literature.

perhaps be strengthened by reference to more recent literature if there is any. Ref 2 is from 2003.		
4. Page 4 Lines 84-87. Are you referring here to primary care, secondary care, or healthcare in general? Might be useful to make this clear here.	Thank you. We have made this clearer.	L84 – we have added ‘healthcare’ – There are many different definitions used in the healthcare literature.
5. Page 5 Lines 101-107. Useful for the rationale but would this sit better elsewhere? There is no connection to the lit here so I think for that reason it might be better placed towards the end of the background.		L117-123 – we have moved this information towards the end of the introduction section.
6. Page 7 Lines 150-152. Embed and respond. Isn't it more likely that you will respond to stakeholder perspectives and then embed those in all stages	In a realist review, stakeholder and PPIE perspectives are embedded throughout all stages of the review process (e.g., development of the initial programme theory, focus of the review, analysis, interpretation of data, and dissemination activities), and responded to accordingly (e.g., PPIE may inform possible ways of thinking about patient care in relation	

of the research?	to access, help-seeking behaviour, and continuity).	
7. Page 7 Line 158. You have as a research question (and research objective on page 6) “to support effective and equitable healthcare outcomes”. I think that is absolutely fine. However, do you discuss health inequalities in the background? It might be worth adding something in the background to support this. It’s an important consideration because it might be that certain organisational practices or arrangements for collaborative and integrated working could potentially privilege certain groups in society so		We have added some background and a reference to further support the focus on effective and equitable healthcare outcomes. L103 – ‘how these impact patient trust, experience, equity, and outcomes’. L115-116 (‘to address local health inequalities’). L104-106 – ‘UK reports in the last decade consistently position CP as a solution for an over-burdened GP and a means to improve access for patients from socioeconomically deprived communities.’ L153-154 – ‘it is less clear how these working arrangements can be made to work well within ICS settings for both healthcare professionals and patients from diverse cultural and socioeconomic backgrounds.’

good objective but some reflection in the background might help strengthen the rationale for that RQ.		
8. Page 7 Line 172. I think you need more clarity and explanation around what you mean by initial programme theory. I just think this needs a bit further explanation particularly for readers less familiar with realist approaches.		We have provided further information. L190-192 – ‘and includes theorising the anticipated interactions between contexts, mechanisms, and outcomes.’
9. Apologies for being pedantic but by PPI do you mean PPIE? Is there Engagement as well as involvement ?	Thank you, we have changed this.	L45-46. L167. L200. L202. L204. L208. L209. L289. L306. L318. We have changed any mention of patient contributors to Patient and Public Involvement and Engagement (PPIE).
10. In your figure outlining your initial programme theory you have GPs, all allied	Realist approaches require the development of an initial programme theory at the start of the review. As we progress with the review, we will have the flexibility to respond to the available	

health professionals, and administrators, and admin staff. Is it worth being a bit more specific here? GP based pharmacists and non-medical prescribers could particularly be involved in collaboration and integration between GP and CP.	literature. We agree that throughout the analysis, clarity regarding the different stakeholders and their roles will be important once we have accumulated and synthesised evidence from the literature.	
11. Page 10 Lines 243-246. Article selection. Could you provide the criteria for deciding if relevant data are credible and trustworthy?		L269-271 – ‘for instance, depending on the type of document, appropriate quality standards will be used in addition to regular research team discussions.’
12. Page 11 Line 252 Data extraction. Discrepancies and disagreements discussed between whom?	We have addressed this – thanks.	L277 – we have added ‘Discrepancies or disagreements will be discussed between the research team in detail and documented.’
13. Page 13 Line 305. You say you will find insights and solutions. Is it beyond the		L331 – We have replaced ‘solutions’ with ‘recommendations’.

scope of a review to provide solutions? You might unpick current practice and develop it into CMOCs and your programme theory but I'm not sure if "solutions" can be offered by that?		
14. Related to that can you perhaps discuss what future empirical work might be undertaken after your review? I also think some comment about the impact beyond the review and what the pathways to that impact might look like would be of use.		L332-334 – 'Importantly, it will help to identify where critical knowledge 'gaps' exist and propose ways to close these gaps, providing direction for future theoretical research and practice.'

VERSION 2 – REVIEW

REVIEWER	Jeffries, Mark University of Manchester, School of Health Sciences
REVIEW RETURNED	28-Nov-2022
GENERAL COMMENTS	Thank you for your amended manuscript. You have addressed all the points I previously made and I am very happy for this now to be accepted for publication.